# ROS-Mediated Nematocidal Activity and Reproductive Toxicity of Herbal Extracts in *Caenorhabditis elegans*

**DOI:** 10.3390/nu17213337

**Published:** 2025-10-23

**Authors:** Anna Hu, Qinghao Meng, Zifei Liu, Yuxuan Wu, Robert P. Borris, Hyun-Min Kim

**Affiliations:** 1Natural and Applied Sciences, Duke Kunshan University, Kunshan 215316, China; 2School of Pharmaceutical Science and Technology, Tianjin University, Tianjin 300072, China

**Keywords:** *Ruscus hyrcanus*, *Juniperus oblonga*, *Stachys lavandulifolia*, DNA repair, meiosis, germline development

## Abstract

Background/Objectives: Traditional medicinal plants are a rich source of phytochemicals with diverse biological effects, yet their safety and mechanistic impact on reproductive health remain underexplored. In this study, we investigated the effects of *Ruscus hyrcanus*, *Juniperus oblonga*, and *Stachys lavandulifolia* extracts on survival, fertility, and germline integrity in *Caenorhabditis elegans*. Methods: Synchronized young adult worms were exposed to each extract, and survival and reproductive parameters were statistically analyzed using two-tailed Mann-Whitney tests. Results: Through LC-MS analysis, we identified that all three extracts shared 78 compounds, mainly including phenolic acids, flavonoids, and terpenoids. Our findings indicate that reactive oxygen species generation is a major driver of nematocidal and fertility-reducing effects, while modulation of DNA damage response pathways further contributes to germline defects. Conclusions: Taken together, these results demonstrate that exposure to the extracts significantly (*p <* 0.05) reduces survival, impairs larval development, elevates the High Incidence of Males phenotype, and disrupts germline integrity in a dose-dependent manner.

## 1. Introduction

Plant-derived compounds have long been recognized as a rich source of bioactive molecules with diverse pharmacological properties. Among these, certain herbs have demonstrated potent effects on nematodes, making them promising candidates for developing environmentally friendly nematocides. *Ruscus hyrcanus*, *Juniperus oblonga*, and *Stachys lavandulifolia* are three such herbs with a history of traditional medicinal use across Europe, the Middle East, and western Asia. Yet their nematocidal potential and underlying mechanisms remain poorly understood.

*R. hyrcanus* is an evergreen shrub endemic to Iran and Azerbaijan, traditionally valued for its diuretic, vasoconstrictive, and antimicrobial properties [1,2,3,4]. *R. hyrcanus* shoots provide essential fatty acids and phytosterols, qualifying it as a functional lipid source [5], yet complete proximate (protein, carbohydrate, mineral, vitamin) data are still lacking in the literature. Its steroidal saponins, flavonoids, and phenolic acids may generate ROS-mediated cytotoxicity, a mechanism reported in related *Ruscus* species [4,6].

*J. oblonga* is a dioecious evergreen tree found in Iran, Turkey, and the Caucasus, with folk applications ranging from diuretic and antiscorbutic to antitumor and neuroprotective activities [7,8]. Recent studies have shown that *Juniperus communis* berry extracts exhibit nematocidal effects in *C. elegans*, causing motility impairment and increased mortality, while essential oils from *Juniperus phoenicea* demonstrated acetylcholinesterase inhibitory activity [9,10]. These findings suggest that *J. oblonga* may exert similar nematocidal effects through disruption of cuticular permeability or interference with neuromuscular signaling.

*S. lavandulifolia*, a herbaceous plant widely used in Turkish and Iranian medicine [11], exhibits analgesic, anti-inflammatory, and antioxidant effects [12,13,14,15]. Traditionally, *S. lavandulifolia* is prepared as a tea or decoction. In line with these bioactivities, its flavonoid-rich extracts show cytotoxicity in mammalian cells and brine shrimp [11], suggesting potential nematocidal effects mediated through oxidative imbalance.

Despite this rich ethnopharmacological knowledge, the mechanistic understanding of these herbs’ effects on nematodes and reproductive biology remains limited. Building on our previous studies, which demonstrated a strong interplay between DNA damage response and repair pathways and nematocidal effects in *Caenorhabditis elegans*, we propose that modulation of genome integrity may be a key determinant of herb-induced toxicity. Guided by this connection, we aimed to investigate the molecular and cellular mechanisms underlying the nematocidal and reproductive effects of these medicinal plants. We therefore hypothesize that common bioactive constituents mediate nematocidal activity against *C. elegans*.

In this study, we investigated the effects of *R. hyrcanus*, *J. oblonga*, and *S. lavandulifolia* extracts on *C. elegans* survival, fertility, and germline DNA damage responses. Using a combination of survival assays, germline cytology, and LC-MS profiling, we examined both the shared ROS-mediated mechanisms and herb-specific contributions to DNA damage and apoptosis. Interestingly, while modulation of DNA damage responses was observed, our findings indicate that ROS generation plays a particularly prominent role in driving the nematocidal and fertility-reducing effects of these herbs. Collectively, this work provides a comprehensive assessment of how traditional medicinal plants and their phytochemicals can modulate oxidative stress and genome stability in vivo. By linking phytochemical composition with ROS-mediated bioactivity and reproductive outcomes, our findings emphasize not only the biological activity but also the safety-relevant cytotoxic potential of these herbs, contributing to a more balanced understanding of their nutritional and therapeutic implications.

## 2. Materials and Methods

### 2.1. Materials and Plant Extracts’ Preparation

All *C. elegans* strains were maintained at 20 °C under standard laboratory conditions, as previously described [16]. The N2 Bristol strain was used as the wild-type reference, and synchronized L1 larvae were generated for all survival and fertility assays. L1 larvae ensures a uniform developmental stage for accurate toxicity assessment. *E. coli* strain OP50 was used as food.

*Ruscus hyrcanus*, *Juniperus oblonga*, and *Stachys lavandulifolia* were collected near Kish Village in the Sheki District of Azerbaijan (GPS coordinates: 41°11′50.91″ N, 47°9′25.65″ E). For *J.o*, roots, stems, leaves, and fruits were individually harvested. *R.h* was collected as a single sample consisting of leaves and fruits, while *S.l* was collected as the entire plant. All plant materials were authenticated, cleaned of extraneous material, air-dried, and ground into coarse powder prior to extraction.

To identify and quantify bioactive compounds in the herbal extracts, we collected the three herbs from Azerbaijan as previously described [17]. Plant material was cleaned, air-dried, and ground into a coarse powder before methanol extraction. A 1 kg portion of each dried sample was sequentially fractionated using n-hexane, dichloromethane, and n-butanol to obtain hexane, butanol, and aqueous fractions. Among them, the butanol fraction was used for all subsequent experiments, as it contained the majority of semi-polar bioactive constituents and was available in sufficient quantity for all analyses. Extracts were dissolved in DMSO (1 mg/mL), stored at −80 °C, and later diluted in M9 buffer to a final concentration of 0.03 µg/mL for analysis. In preliminary tests, 0.01% (*v/v*) butanol and hexane showed no observable effects on *C. elegans* survival. DMSO (≤0.1%) was used as the vehicle control in all subsequent experiments.

### 2.2. LC-MS Analysis

LC-MS profiling was performed to identify and quantify bioactive constituents in the herbal extracts, using a Shimadzu LC-30A system coupled to an AB Sciex Triple TOF 5600+ mass spectrometer (YanBo Times, Beijing, China, [18]). Chromatography was conducted on a C18 column (2.2 μm, 2.1 × 100 mm) at 40 °C with a flow rate of 0.2 mL/min. Gradient elution employed 0.1% formic acid in water (A) and acetonitrile (B), with a linear gradient from 5–95% B over 30 min. MS detection used electrospray ionization in both positive and negative modes, with optimized collision energy and declustering potential for MS/MS fragmentation.

Because this analysis followed an untargeted LC-MS profiling approach, no authentic standards were employed, and compound identification was based on accurate mass and fragmentation patterns. In addition, some compound names were translated from the original Chinese names provided by the instrument software.

### 2.3. Antibacterial Activity

The impact of herb extracts on the proliferation of *E. coli* OP50 was assessed by measuring optical density at 600 nm, following the methodology outlined in [17]. Each extract (*R. h*, *J. o*, or *S. l*) was tested at a concentration of 0.03 µg/mL to evaluate potential antibacterial activity by tracking bacterial growth over time. Briefly, the bacterial growth assay was performed in 15 mL tubes (Falcon, Noble Park North, Australia, 352095). 500 µL of an OP50 overnight culture was inoculated into 10 mL of LB medium and maintained overnight at 37 °C with gentle shaking (100 RPM). On the following day, 100 µL of the culture was transferred to a 96-well plate, and OD_600_ values were recorded at regular intervals over 24 h. DMSO (0.1%) served as the vehicle control. OD_600_ values were recorded from 0–24 h.

### 2.4. In Vivo Studies on C. elegans

To assess nematocidal and reproductive toxicity of the herbal extracts, synchronized L1 larvae were obtained from gravid hermaphrodites on NGM plates following established protocols [19].

Larvae were incubated in herb extract solutions in 96-well plates at 20 °C with gentle agitation. Survival and mobility were assessed after 24 h. Larval arrest/lethality was defined as the percentage of hatched worms failing to reach adulthood, and male proportion (Him) as the percentage of adult males. Controls: untreated worms in DMSO vehicle were used. Statistical significance was evaluated using two-tailed Mann-Whitney tests with 95% confidence intervals. Experiments were performed in triplicate [20].

### 2.5. Dose-Dependent Nematocidal Effects of the Extracts

To validate nematocidal activity, synchronized L1 larvae were incubated with increasing concentrations (0.03, 0.3, and 3 µg/mL) of herbal extracts from *Ruscus hyrcanus* (*R.h*), *Juniperus oblonga* (*J.o*), and *Stachys lavandulifolia* (*S.l*) in 96-well plates at 20 °C with gentle agitation. Controls consisted of worms treated with DMSO vehicle. Survival and mobility were assessed after 24 h. Dose-dependent effects were evaluated by comparing survival rates across the different extract concentrations. Statistical significance was determined using two-tailed Mann-Whitney tests with 95% confidence intervals.

### 2.6. Germline Integrity Analysis by Immunofluorescence

To visualize germline DNA damage and checkpoint activation, whole-mount gonads were processed for immunofluorescence staining according to previously established protocols [19]. The primary antibody used was rabbit anti-pCHK-1 (1:250, Cell Signaling, Ser345, Danvers, MA, USA), and detection was performed with Cy3-conjugated anti-rabbit secondary antibodies (1:300, Jackson Immunochemicals, West Grove, PA, USA). Images were captured using a Nikon Eclipse Ti2-E software (Nikon, Tokyo, Japan) inverted microscope equipped with a DSQi2 camera (Nikon, Tokyo, Japan), with optical sections collected at 0.2 μm intervals. A 60× objective combined with 1.5× auxiliary magnification was used, and deconvolution was applied via NIS Elements software AR (Nikon, Tokyo, Japan). Partial projections of half-nuclei are displayed. All reactions were performed at least in duplicate.

### 2.7. Measurement of Brood Size

To assess nematocidal and reproductive toxicity of the herbal extracts, synchronized L1 larvae were obtained from gravid hermaphrodites on NGM plates following established protocols [19]. Larvae were incubated in herb extract solutions in 96-well plates at 20 °C with gentle agitation. Brood size was measured over 4–5 days post-L4 stage by transferring the worms to a new plate each day. Controls: untreated worms in DMSO vehicle were used. Statistical significance was evaluated using two-tailed Mann-Whitney tests with 95% confidence intervals. Experiments were performed in triplicate [20].

### 2.8. Quantification of DNA Damage Response and Apoptosis in the Germline

To quantify checkpoint activation and germline apoptosis, pCHK-1 foci were quantified as described in [21,22], analyzing five to ten gonads per treatment. Germline apoptosis was evaluated using acridine orange staining in age-matched worms at 20 h post-L4 stage, following [23]. Between 20 and 30 gonads per condition were scored using a Nikon Ti2-E fluorescence microscope. Significance was assessed with a two-tailed Mann-Whitney test at a 95% confidence interval. All reactions were performed at least in duplicate.

To measure transcriptional changes in specific genes, RNA was extracted from young hermaphrodites, and first-strand cDNA synthesis was performed using the ABscript II First Strand Kit (ABclonal, RK20400, Shanghai, China). qPCR reactions were carried out with ABclonal 2X SYBR Green Fast Mix (RK21200) on a LineGene 4800 system (BIOER, FQD48A, Beijing, China). The cycling protocol consisted of an initial 95 °C denaturation for 2 min, followed by 40 cycles of 95 °C for 15 s and 60 °C for 20 s. A melting curve analysis from 60 °C to 95 °C verified amplicon specificity. The tubulin gene *tba-1* served as an internal reference based on *C. elegans* microarray data [24,25]. All reactions were performed at least in duplicate.

### 2.9. ROS Measurement and Statistical Analysis

To determine whether the nematocidal activity of the herbal extracts was mediated by reactive oxygen species (ROS), *C. elegans* were co-treated with the ROS scavenger N-acetyl-L-cysteine (NAC, Sigma-Aldrich A9165, Burlington, MA, USA [26]. Synchronized L4-stage worms (n = 30 per treatment) were transferred to NGM plates supplemented with each herbal extract at the indicated concentrations. NAC was freshly prepared in M9 buffer and added to achieve final concentrations of 1 mM or 2 mM. Worms were incubated with herbal extracts with or without NAC at 20 °C for 24 h. Following incubation, worm survival was scored under a dissecting microscope. Worms failing to respond to gentle touch with a platinum wire were considered dead. Survival rates were determined from three independent biological replicates.

All quantitative data were analyzed using GraphPad Prism (version 9.5, GraphPad Software, San Diego, CA, USA) and Instat (version 3.1, GraphPad Software, San Diego, CA, USA). All statistical tests in this report were performed using two-tailed Mann-Whitney tests unless otherwise indicated. A *p*-value < 0.05 was considered statistically significant. All experiments were performed in at least three independent biological replicates, and data are presented as mean ± standard error of the mean (SEM) unless otherwise indicated.

## 3. Results

### 3.1. Yield of the Herb Extract

One kilogram of air-dried, ground material from each plant (*Ruscus hyrcanus*, *Juniperus oblonga*, *Stachys lavandulifolia*) was sequentially partitioned with n-hexane, dichloromethane and n-butanol. After solvent removal, the resulting n-butanol residues were dissolved in DMSO to give a stock concentration of 1 mg/mL and stored at −80 °C; the exact dry weights of the residues are unknown, so percentage yields cannot be reported. The *Ruscus hyrcanus* extract (RHE), *Juniperus oblonga* extract (JOE), and *Stachys lavandulifolia* extract (SLE) were used in all analyses.

### 3.2. LC-MS Profiling of Three Herbal Extracts Reveals 79 Bioactive Compounds, Including 13 with Nematocidal Potential

LC-MS analysis was performed to identify and quantify the major bioactive constituents present in the herbal extracts (Figure 1). All three n-butanol fractions of *Ruscus hyrcanus* extract (RHE), *Juniperus oblonga* extract (JOE), and *Stachys lavandulifolia* extract (SLE) revealed 79 compounds, 78 of which were common, although their relative abundances varied among the extracts (Appendix A).

#### Four Major Categories

LC-MS profiling revealed four major classes of metabolites: phenolic compounds, glycosides, isoprenoids, and lipid (Table 1 and Appendix A).

### 3.3. Antibacterial Activity on E. coli OP50

Because the decreased survival of *C. elegans* following exposure to the three herbal extracts could potentially arise from indirect effects on their bacterial food source, we tested whether the extracts inhibited the growth of *E. coli* OP50. Cultures of OP50 were incubated with each extract for 24 h, and bacterial proliferation was monitored by optical density at 600 nm. No significant differences were observed between the extract-treated groups and the DMSO control at either 12 or 24 h, indicating that the extracts did not interfere with bacterial growth under the tested conditions. At the 24 h time point, OD values remained nearly identical across groups (Figure 2, 0.12 for OP50 + DMSO vs. 0.13 for OP50 + RHE, JOE or SLE; *p =* 0.076, 0.1000, 0.1000, respectively). These results demonstrate that the effects of the herbal extracts are not attributable to altered bacterial food availability.

### 3.4. In Vivo Studies on C. elegans

Extracts from all three herbs—*Ruscus hyrcanus (R.h)*, *Juniperus oblonga (J.o)*, and *Stachys lavandulifolia (S.l)*—significantly reduced the survival rates of *C. elegans* compared with the untreated control group. The extent of reduction was comparable to that observed in the DMSO control (Figure 3, 92.7 vs. 60.3 for +DMSO and *R.h*; 92.7 vs. 38.5 for +DMSO and *J.o*; 92.7 vs. 52.9 for +DMSO and *S.l* at 0.03 µg/mL). This suggests that all three extracts contain phytochemical compounds detrimental to the health and viability of *C. elegans*. In addition, exposure to the extracts led to larval arrest and lethality (92.7 vs. 60.3 adult (%) for +DMSO and *R.h*; 100 vs. 40.1 for +DMSO and *J.o*; 100 vs. 60.9 for +DMSO and *S. l* at 0.03 µg/mL). Together, these findings indicate that the three herbs exert potent nematocidal activity, likely linked to growth defects induced by bioactive compounds within the extracts.

In *C. elegans*, errors in sex chromosome segregation during meiosis often result in an increased occurrence of males, a condition referred to as the High Incidence of Males (HIM) phenotype. The HIM phenotype is a well-established marker for studying the effects of environmental or chemical factors on reproductive health and genetic stability [27]. Exposure to the three herbal extracts elevated the HIM phenotype, with the strongest effect observed in *S.l*, suggesting that this extract may interfere with proper sex chromosome segregation (Figure 3, 2.8 vs. 3.63 for +DMSO and *R.h*; 2.8 vs. 5.2 for +DMSO and *J.o*; 2.8 vs. 17.9 for +DMSO and *S.l* at 0.03 µg/mL).

The increased HIM phenotype, particularly prominent with *S.l* extract, implies that these herbs may induce underlying genetic instability in *C. elegans*. Disruption of normal meiotic processes leading to chromosome mis-segregation is a hallmark of genomic instability and is associated with various diseases, including cancer [19]. These results highlight the potential of the three herbs as sources of nematocidal compounds with possible implications for genomic stability.

### 3.5. Dose-Dependent Nematocidal Effects of the Extracts

To further validate the nematocidal effects, we investigated whether different concentrations of the extracts correlated with changes in *C. elegans* phenotypes. Increasing concentrations of the herbal extracts consistently led to reduced survivability, demonstrating a clear dose-dependent relationship (Figure 3). For example, in *R.h*-treated worms, survival decreased from 60.3% to 57.2% and 55.1% at concentrations of 0.03, 0.3, and 3 µg/mL, respectively (*p <* 0.05 compared with control). Similarly, survival rates for *J.o* declined from 38.5% to 35.5% and 33.4% (*p <* 0.05), while those for *S.l* decreased from 52.9% to 49.8% and 47.7% (*p <* 0.05) across the same concentration range. These findings confirm a strong correlation between extract concentration and worm mortality, supporting the hypothesis that higher doses of these herbal extracts significantly compromise nematode viability.

Taken together, extracts from the three medicinal plants—*Ruscus hyrcanus*, *Juniperus oblonga*, and *Stachys lavandulifolia*—exhibited strong nematocidal activity in *C. elegans*, causing dose-dependent reductions in survival and larval development. These effects occurred independently of bacterial growth. Exposure to the extracts also increased the High Incidence of Males (HIM) phenotype, particularly with *S. lavandulifolia*, suggesting potential disruption of sex chromosome segregation during meiosis.

### 3.6. Effects of Herbal Extracts on Germline Nuclear Organization and Integrity

In *C. elegans*, the development of the germline is a tightly regulated process where nuclei follow a specific spatial and temporal arrangement as they progress through various stages of meiosis and mitosis. During normal germline progression, mitotic nuclei are located at the distal tip of the premeiotic zone (PMT) and gradually transition to meiotic prophase as they move towards the transition zone (TZ), where nuclei begin to adopt a crescent shape. This shift marks the commencement of the meiotic process [28]. Given that exposure to *S.l*. extracts resulted in defective meiotic progression in our initial observations, we aimed to further investigate how these herbal extracts affect the overall organization of the germline in *C. elegans*.

#### 3.6.1. Effects of Herbal Extracts on Germline Nuclear Spacing

To explore this, we dissected adult hermaphrodites and examined the arrangement of nuclei in the germline. For this, we quantify the distances between adjacent nuclei. While all three herb extracts treatment did not alter the distance in PMT or TZ significantly. *S.l*-exposed worms showed significant distance between nuclei at pachytene (Figure 4A,B, and Appendix A). Specifically, in the pachytene stage, the gap between nuclei decreased from 16 µm in the control group to 10.6 µm in the *S.l*-treated group (*p <* 0.0001), suggesting mild but significant defective meiotic progression together with robust HIM phenotype.

#### 3.6.2. Germline Nuclear Morphology and Mitotic Integrity

Another typical feature of *C. elegans* germline is the appearance of crescent-shaped nuclei, which marks the transition from mitosis to meiosis [29]. However, crescent-shaped nuclei represented in PMT or pachytene often indicate abnormal development of germline nuclei. Three herb extracts did not alter the frequency of crescent-shaped nuclei in the PMT, TZ and pachytene stages. In the PMT stage, all three herb and control group exhibited 0–0.7 crescent shape nuclei per gonad arm (Figure 4C and Appendix A). Similarly, no significant changes were visible at TZ and Pachytene.

Formation of chromatin bridges between gut cells is a hallmark of defective chromosomal segregation during anaphase or cytokinesis, often found in exogenous DNA damage [18,24], however, no such chromatin bridges were observed in any of the three herb-treated worms (Figure 4D). Under identical imaging conditions, *Torenia* sp. extract clearly exhibited chromatin bridges [18], confirming that the assay can resolve such structures when present. This indicates that the disruption caused by these three herb extracts may not primarily manifest as mitotic defects in gut cells or other mitotic tissues.

### 3.7. Three Herb Extracts Significantly Reduce Brood Size in C. elegans

Given the reduction in survival observed in *C. elegans* exposed to three herb extracts, we investigated whether these defects in germline progression might also lead to a decrease in fertility. For this, we quantified the brood size of hermaphrodites exposed to *three herb* extracts over a period of four days starting from the L4 stage. Compared with control worms, the three herb extract-treated worms exhibited a significant drop in fertility starting on day 1 and continued until day 3 or 4 (Figure 4E). Consistently, total brood sizes decreased from 156 in control to 140, 137, and 122 in *R.h*, *J.o,* and *S.l*, respectively (Appendix A). This reduction in fertility further supports the idea that these herb extracts interfere with meiotic development, resulting in compromised reproductive success.

### 3.8. S.l Extract, but Not R.h or J.o, Modulates the DNA Damage Checkpoint Pathway Through pCHK-1

The DNA damage response constitutes a critical signaling cascade that initiates processes such as DNA repair, apoptosis, and cell cycle arrest [21,30,31]. Both ATM and ATR can activate CHK1 directly or via intermediary kinases, with CHK1 activation driving downstream responses that facilitate DNA repair, arrest cell cycle progression, and preserve genome integrity in response to DNA damage or replication stress.

We examined the expression levels of *atm-1* (mammalian ATM), *atl-1* (mammalian ATR), and pCHK-1 (homologous to mammalian phospho-CHK1). Treatment with *R.h* and *J.o* herb extracts did not alter the expression of these two key DNA damage checkpoint components, suggesting that these treatments do not activate the DNA damage response. In contrast, *S.l* extract mildly but significantly reduced *atm-1* expression (Figure 5A, 1.0 vs. 0.76 in control and *S.l*, *p =* 0.012), whereas *atl-1* expression remained unchanged (1.0 vs. 0.98, *p =* 0.3947). This finding suggests that *S.l* extract may selectively attenuate ATM-mediated DNA damage signaling without broadly impairing ATR-dependent pathways, thereby indicating a potential herb-specific modulation of genome surveillance mechanisms rather than a general suppression of the DNA damage response.

Consistently, the downstream target of ATM and ATR, pCHK-1, did not show increased levels in the pachytene stage of germlines upon *R.h* or *J.o* treatment (Figure 5B and Appendix A). However, *S.l* treatment significantly induced pCHK-1 foci levels (*p =* 0.0326 for control vs. *S.l*).

In *C. elegans* hermaphrodites, germ cells undergo apoptosis in response to environmental stresses such as DNA damage. Germ cell apoptosis induced by DNA damage requires the DNA damage checkpoint, *cep-1* and *egl-1*, but not the pathway for physiological apoptosis [22,32,33]. We therefore analyzed DNA damage-induced apoptosis in the germline. Compared to the untreated control group, treatment with *R.h*, or *J.o* did not alter apoptosis levels during the pachytene stage (Figure 5C and Appendix A). However, consistent with the pCHK-1 and *atm-1* expression results, *S.l* exposure significantly increased apoptosis levels (0.9 vs. 1.5 in control and *S.l*).

Together, our apoptosis data indicate that exposure to *S.l* extract modestly but significantly altered the DNA damage checkpoint pathway and subsequent apoptosis, whereas *R.h* and *J.o* extracts did not affect these pathways.

### 3.9. ROS Contributes to the Nematocidal Action of Herbal Extracts

Given that several of these compounds exert their bioactivities through ROS generation, and that ROS is also known to contribute to insecticidal activity, we examined whether ROS contributes to nematocidal death. To test this, we co-treated *C. elegans* with each extract and the ROS scavenger N-acetyl-L-cysteine (NAC) and measured survival rates. All three extracts significantly reduced nematode survival compared to the untreated control (Figure 6, *R.h*: 69.6%, *J.o*: 42.6%, *S.l*: 52.6%; *p <* 0.001, n = 30). Co-administration of NAC restored survival in a concentration-dependent manner: at 1 mM NAC, survival rose to 81.8% (*p* = 0.0160) for *R.h*, 59.4% (*p* = 0.0120) for *J.o*, and 80.6% (*p* = 0.0095) for *S.l*; at 2 mM, it further increased to 85.0% (*p* = 0.0079), 71.8% (*p* = 0.0117), and 89.8% (*p* = 0.0093), respectively. These results demonstrate that the nematocidal effect of all three herbs is, at least in part, mediated through ROS-dependent cytotoxicity.

Taken together, our compound categorization revealed that multiple constituents from the three herbal extracts possess nematocidal or related bioactivities, frequently mediated through ROS generation.

## 4. Discussion

In our previous screening of over 316 herb extracts, ~16% caused reduced survival, larval arrest, or lethality in *C. elegans* [17]. Three extracts showing the most consistent effects were selected for focused study. *C. elegans* was chosen for its well-characterized development, short lifecycle, fully sequenced genome, and reproducible endpoints, allowing rapid, high-throughput evaluation of nematocidal effects. Its conserved stress response and ROS-mediated pathways also make it suitable for investigating herb-induced cytotoxicity.

### 4.1. Chemical Composition as a Basis for Functional Diversity

LC-MS profiling of the three herb extracts revealed both shared and unique chemical features. Among them, 78 compounds were common to all extracts, including 13 known or predicted to have bioactivity. These shared bioactive compounds likely account for the extracts’ conserved ROS-dependent effects. However, differences in the relative abundance of these constituents may underlie the herb-specific activities observed in SLE versus the primarily ROS-driven responses of RHE and JOE.

Both *Stachys* and *Juniperus* species have been reported to exhibit nematocidal activity. In *Juniperus*, leaf and berry extracts as well as essential oils were shown to inhibit nematode survival and development [10,34,35]. In *Stachys*, hydroethanolic extracts of *S. sylvatica* reduced the viability of *Rhabditis* nematodes, with higher concentrations showing effects comparable to albendazole [36], while alcohol extracts of *S. recta* induced nearly 47% mortality after 24 h of exposure [37]. LC-MS analysis of *S. sylvatica* revealed 17 major bioactive metabolites, including chlorogenic acid and its isomers, flavonoid glycosides, and verbascoside derivatives [36].

To further interpret these findings, we categorized the detected compounds based on their reported nematocidal or related bioactivities (Figure 1). Among the terpenoids, menthyl acetate displayed contact toxicity and repellency against Lasioderma serricorne adults [38]. Tanshinone II induced cardiotoxicity and developmental malformations in zebrafish embryos [39], while emodin exhibited insecticidal activity against Nilarparvata lugens and Mythimna separata [40]. Tormentic acid caused apoptosis and G0/G1 cell cycle arrest in MCF-7 cells, mediated by reactive oxygen species (ROS) and mitochondrial dysfunction [41].

Flavonoid derivatives also demonstrated nematocidal or parasiticidal effects through multiple mechanisms. Naringenin enhanced lifespan and reduced ROS accumulation in *C. elegans* [42], although it produced sublethal developmental effects in amphibian embryos [43]. Hesperidin showed schistosomicidal activity in vitro, achieving complete mortality of Schistosoma mansoni adults [44]. Apigenin inhibited larval growth in *C. elegans* through activation of the DAF-16 pathway [45], and quercetin derivatives caused neuromuscular paralysis via ROS-mediated effects, ultimately leading to paralysis and death of worms [46]. Other compounds, such as 7-Hydroxycoumarin showed cytostatic and apoptotic effects in several human cancer cell lines [47], and palmitic acid induced oxidative stress, endoplasmic reticulum and mitochondrial dysfunction, and apoptosis in Chang liver cells [48].

In contrast to the nematocidal compounds, a few flavonoid derivatives primarily exhibited protective or stress-tolerance effects rather than direct toxicity. Icariin and its metabolic derivative icariside II, extended lifespan in *C. elegans* [49]. Treatment with icariside II enhanced thermotolerance and oxidative stress resistance, slowed locomotion decline in late adulthood, and delayed the onset of paralysis caused by polyQ and Aβ1–42 proteotoxicity in an insulin/IGF-1 signaling (IIS) pathway-dependent manner. Similarly, luteolin derivatives did not alter the lifespan of wild-type N2 worms under normal conditions but conferred protective effects under oxidative stress [50].

Although quantitative correlations could not be established due to the qualitative nature of the LC-MS data, our comparative analysis indicates that extracts enriched in polyphenolic or terpenoid compounds exhibit stronger nematocidal and ROS-inducing activities. This observation is consistent with previous reports describing the nematocidal properties of these phytochemical classes [51]. It should also be noted that the extracts were tested in their unfractionated form; therefore, the observed biological effects likely result from the combined action of multiple constituents rather than a single dominant compound. This complexity underscores the need for future fractionation and compound-specific validation to disentangle the individual and synergistic contributions underlying the overall bioactivity. These findings provide a basis for investigating the biological effects of these extracts in *C. elegans*.

### 4.2. Nematocidal and Fertility-Reducing Effects

This study demonstrates RHE, JOE, and SLE exert pronounced biological activity in *C. elegans*, manifested as reduced survival, developmental arrest, and markedly decreased reproductive output. For *J. o*, roots, stems, leaves, and fruits were individually harvested, whereas *R.h* was collected as a single sample consisting of leaves and fruits, and *S. l* was collected as the entire plant. The choice of plant parts and solvent fractions was not predetermined but guided by preliminary *C. elegans* assays, which identified the n-butanol extracts as those showing the most reproducible bioactivity among the tested fractions.

All plant materials were authenticated, cleaned of extraneous material, air-dried, and ground into coarse powder prior to extraction. The n-butanol fraction was specifically chosen to enrich for mid-polar phytochemicals, consistent with prior reports showing higher biological activity of such fractions.

Beyond these outcomes, the extracts perturbed meiotic progression and increased the HIM phenotype, with SLE producing the strongest impact. These findings suggest that phytochemicals within these herbs influence germline integrity, and that fertility reduction reflects not only general toxicity but also impaired meiotic fidelity. Interestingly, these changes occurred without activation of canonical DNA damage checkpoints in RHE and JOE, raising the possibility that their influence on genome surveillance proceeds through non-canonical or herb-specific mechanisms, highlighting the importance of considering both bioactivity and potential safety concerns.

### 4.3. Herb-Specific Differences in DNA Damage Responses

Although ROS-dependent cytotoxicity was a shared feature across all three extracts, they exhibited some herb-specific effects on genome stability. Notably, SLE showed unique changes in markers associated with DNA repair and apoptosis, suggesting a secondary, herb-specific modulation of genome maintenance. In contrast, RHE and JOE induced strong lethality and fertility reduction without detectable canonical checkpoint activation. These observations indicate that while ROS-mediated effects represent the central mechanism driving bioactivity, certain phytochemicals may additionally influence genome regulation in a herb-specific manner.

Although the three extracts share 78 identified compounds, a more accurate interpretation is that they contain largely overlapping chemical classes. Many compounds were detected across all three herbs, yet their relative abundances varied, and each extract possessed certain unique constituents.

### 4.4. ROS-Dependent Cytotoxicity as a Common Mechanism

This overlap at the level of chemical families may account for shared biological effects, although it does not necessarily imply identical composition or activity. Consistent with these observations, prior studies have reported ROS-dependent lethality of quercetin and emodin derivatives in nematodes and insects, confirming that the present herb extracts act through conserved oxidative mechanisms [40,46].

Our functional assays suggest that ROS generation may represent a unifying mechanism underlying the biological effects of all three herbs. Co-treatment with the ROS scavenger NAC partially restored survival in a dose-dependent manner, indicating that ROS-dependent cytotoxicity likely contributes to much of the lethality and fertility reduction. These findings support a model in which oxidative stress acts as a central mediator of herb-induced toxicity, independent of bacterial food supply, and position ROS as an important determinant of physiological outcomes. Importantly, this highlights that phytochemicals can act as a double-edged sword—exerting potent nematocidal activity through ROS generation, while also causing cytotoxic effects that may compromise genome stability, an issue relevant to safety assessment.

## 5. Conclusions

Taken together, our results reveal a dual framework of action. First, all three herbal extracts reduced survival and fertility in *C. elegans* in a dose-dependent manner, with brood size decreasing by up to ~22% and the High Incidence of Males (HIM) phenotype increasing by ~17.9% at the highest concentrations tested. ROS-dependent cytotoxicity was confirmed as a central mechanism, as co-treatment with the ROS scavenger NAC partially restored survival and reproductive output. Second, herb-specific modulation of genome stability was observed: SLE uniquely increased pCHK-1 foci per pachytene-stage nucleus and altered apoptosis levels, indicating perturbation of DNA repair and checkpoint responses. These findings highlight that both qualitative and quantitative variation in phytochemical composition shapes biological outcomes. Future studies should aim to isolate the active constituents, quantify their relative contributions, and determine how their interactions balance beneficial versus adverse effects. By linking phytochemical profiles with observed bioactivity and safety-relevant outcomes, our study provides a data-driven foundation for evaluating traditional medicinal plants in the context of nutraceutical potential and toxicological risk.

## Figures and Tables

**Figure 1 nutrients-17-03337-f001:**
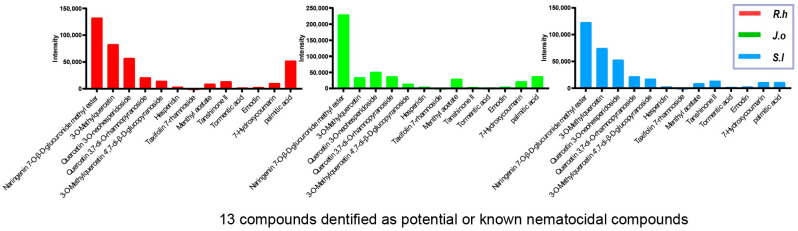
LC-MS profiling of shared nematocidal compounds in three herb extracts. LC-MS analysis identified 78 compounds common *to Ruscus hyrcanus* extract (RHE), *Juniperus oblonga* extract (JOE), and *Stachys lavandulifolia* extract (SLE), among which 13 compounds are known or predicted to have nematocidal activity. The peak intensities of these 13 compounds are plotted on the *y*-axis for each herb extract.

**Figure 2 nutrients-17-03337-f002:**
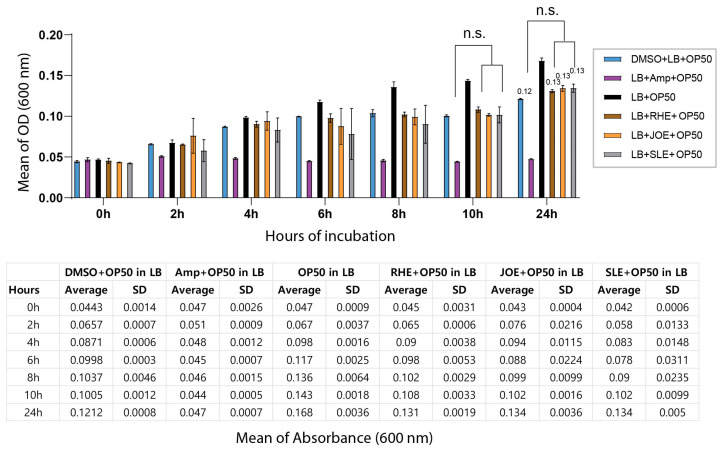
Herb extracts did not affect the growth of *E. coli* OP50. Bacterial growth was monitored over a 24 h period by measuring optical density at 600 nm (OD_600_). The top panel shows the growth curves of OP50 from 0–24 h, and the bottom panel presents the corresponding OD_600_ values. ‘n.s.’ indicates no significant difference compared to the control.

**Figure 3 nutrients-17-03337-f003:**
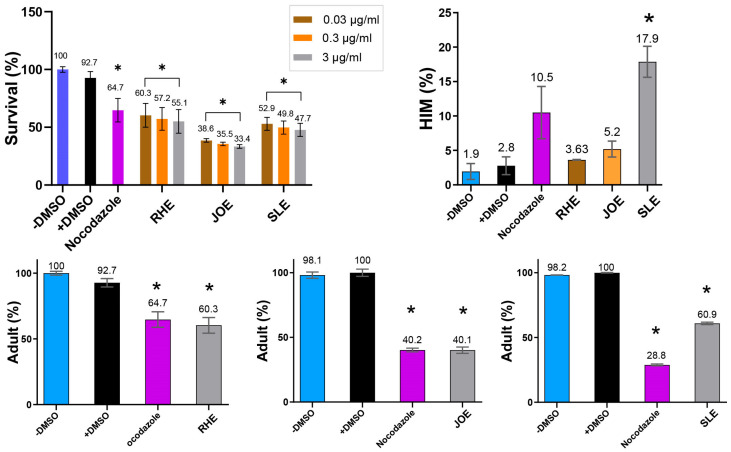
Dose-dependent nematocidal activity of three herb extracts. Survival of *C. elegans* decreased with increasing extract concentration (0.03–3 µg/mL), revealing potent, dose-responsive toxicity. High-incidence-of-male (HIM, %) increased with extracts, while adult proportion (%) decreased. Vehicle control: 0.42 μM DMSO; positive control: 2.7 μM nocodazole for survival, HIM and adults (RHE); 1.4 μM nocodazole for adults (JOE and SLE). * indicates *p <* 0.05 compared with control.

**Figure 4 nutrients-17-03337-f004:**
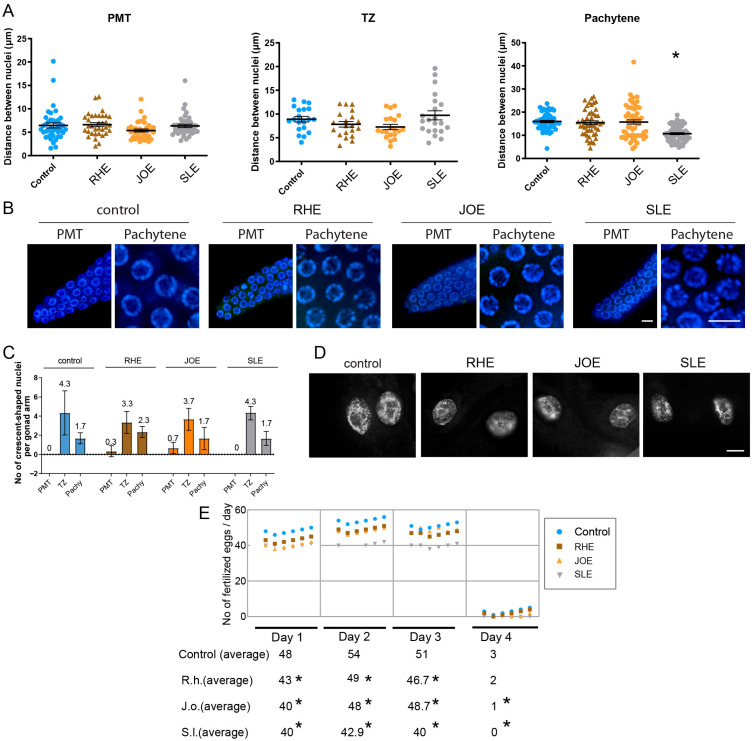
Herb-induced meiotic and fertility defects. (**A**) Nuclear spacing in PMT, TZ and pachytene was unchanged in RHE and JOE; only SLE induced a significant decrease in pachytene nuclear gap (*p <* 0.0001, Mann-Whitney test). (**B**) Representative images of germline nuclei at PMT (premeiotic tip) and pachytene stages upon herb treatment. Bar = 2 µm. (**C**) Frequency of crescent-shaped nuclei remained unchanged across PMT, TZ, and pachytene stages (Mann-Whitney test). (**D**) No chromatin bridges were detected in any herb-treated gut cells, indicating absence of distinct mitotic segregation defects. Bar = 10 µm. (**E**) Brood size was significantly reduced from day 1 to day 3 in all herb groups, confirming compromised fertility without mitotic disruption. Asterisks denote significance versus control group.

**Figure 5 nutrients-17-03337-f005:**
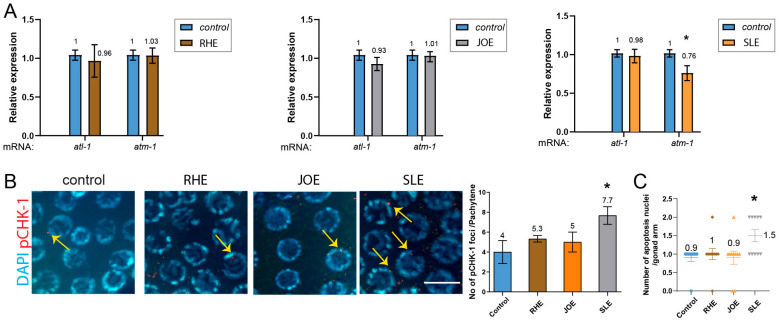
SLE down-regulates *atm-1* and triggers *pchk-1* foci and apoptosis, while RHE and JOE leave all DNA-damage checkpoint and apoptotic markers unchanged. (**A**) qPCR analysis of *atm-1* and *atl-1* expression. RHE and JOE did not alter transcript levels, whereas SLE significantly reduced *atm-1* expression. (**B**) pCHK-1 foci in pachytene-stage nuclei. RHE and JOE showed no change compared to controls (n.s.), while SLE treatment significantly increased pCHK-1 foci (arrows). pCHK-1 was detected with rabbit anti-pCHK-1 primary antibody followed by Cy3-conjugated anti-rabbit secondary antibodies (red channel). DAPI was used for DNA (blue). Bar = 2 µm. (**C**) Germline apoptosis. RHE and JOE showed no change, whereas SLE significantly elevated apoptosis (mean = 1.5; gray inverted triangles indicate individual counts: 5 gonads with 1 corpse and 5 gonads with 2 corpses). Asterisks denote significance versus control group.

**Figure 6 nutrients-17-03337-f006:**
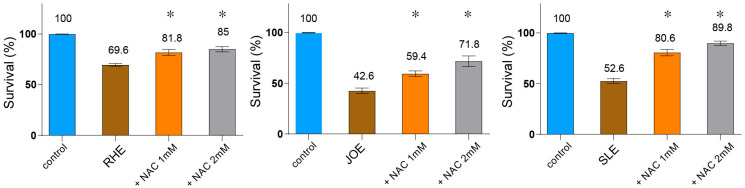
Nematocidal activity of three herb extracts is mediated via ROS-dependent cytotoxicity. *C. elegans* were co-treated with each extract and the ROS scavenger N-acetyl-L-cysteine (NAC), and survival was recorded. All three herb extracts (each used at 0.03 µg/mL) significantly reduced survival compared to the untreated control (n = 30). Co-administration of NAC restored survival in a concentration-dependent manner (1 mM and 2 mM), demonstrating that the nematocidal effect is, at least in part, mediated through ROS-dependent cytotoxicity. Asterisks indicate statistical significance versus the herb-only group.

**Table 1 nutrients-17-03337-t001:** Categorical grouping of the LC-MS-identified compounds shared *Ruscus hyrcanus* extract (RHE), *Juniperus oblonga* extract (JOE), and *Stachys lavandulifolia* extract (SLE). 79 compounds are present in the three extracts and are classified here into four major chemical families: Phenolic compounds, Glycosides, Isoprenoids, and Lipids & derivatives. Many of these metabolites, including flavonoids, terpenoids, and anthraquinones, are known or predicted to exhibit nematocidal, antioxidant, or ROS-modulating activities. Each subclass is represented by selected compounds to illustrate the chemical diversity that may underlie the shared bioactivity of the herbal extracts. Detailed compound lists and corresponding LC-MS spectra are provided in Appendix A.

Types of Compounds	Subclass	Compound 1	Compound 2
**Phenolic compounds**	Flavonoids	Kaempferol	Quercetin 3-O-neohesperidoside
	Coumarins	7-Hydroxycoumarin	8-(Dimethylallyl)-7-hydroxy-6-methoxycoumarin
	Lignans	Syringaresinol	Gomisin D
	Anthraquinones	Emodin	
	Phenols	3,4-Dimethoxyphenyl-β-D-glucopyranoside	
	Polyphenols	Salvianolic acid F	Salvianolic acid K
**Glycosides**	flavonoid glycoside	Taxifolin 7-O-rhamnoside	Vitexin-4″-O-glucoside
	coumarin glycoside	Baihuaqianhuaside	
	Iridoids glycoside	Harpagide	Geniposide
**Isoprenoids**	Terpenes	Tormentic acid	
	Terpenoids	Tanshinone IIA	Artemisinic aldehyde
**Lipids & derivatives**	Fatty acids	Palmitic acid	Isostearic acid

## Data Availability

Data are contained within the article and Appendix A.

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
