# Peer review of "ROS-Mediated Nematocidal Activity and Reproductive Toxicity of Herbal Extracts in Caenorhabditis elegans"

_nutrients, 2025, doi:10.3390/nu17213337_

Round 1

Reviewer 1 Report

Comments and Suggestions for Authors

The manuscript by Hu and collaborators is very interesting since it investigates the possibility to find nematocidal compounds among the extract of three medicinal plants, widely used in the traditional medicine of middle east countries. 

The authors have used leaves and stalks from the three plants and proceeded to three extractions: water, hexane and butanol. They have decided to characterise only the butanol fraction by LC-MS, though without any further explanation. All the compounds are presented in the supplementary table, unfortunately several compounds have only the Chinese ideogram name (11, 12, 34, 50, 54, 69, 102, 111, 92, 94), sometimes the same compound is present in two or more lines (ex. artemisinine 95 and 100), sometimes with the same MW, sometimes with diverse molecular mass and atomic composition.

What is more important is that the authors claim throughout the manuscript that the list of compounds is the same across the 3 plant extracts, but this is not true. In the supplementary table the total number of compounds is the same, but not their composition. Moreover, the total number is linked to the type and length of the column used to perform the chromatography step ahead of the mass spectrometry analysis, this could not be an intrinsic property of the three plants.

The classes of compounds are in common, but the fact that a common chemical scaffold is conserved does not mean that the biological effect is the same. Moreover, there are discrepancies between the content in the list (supplementary table) and Figure 4A and 4B.

In figure 4A the reader thinks that the peak that elutes at a given position is the same molecule in the three plants, but this is not at all true. Example: 3-O-methylquercetine is the peak no. 44 in Rh, 47 in J.o. and 46 in S.l.

Moreover, the total elution times of the three LC-MS are not the same: they go from 1.02 to 31.52 min for R.h, from 1 min to 31.55 min for J.o., from 1.02 min to 33.56 min for S.l. However, if one looks at the peaks with the same elution times, only some compounds are in common (ex. 3-O-methylquercetine exits at 1.28 min in the three herbal extracts), but not all.

The compounds that are "found more than 2 times" come from the same peak with the same retention time, the same intensity and the same chemical formula. This is just 1 repeated line, not two compounds. The case of Naringenin-7-O-beta-D-glucuronide-methylester is peculiar: a compound with formula C22H22O11 is repeated four times (2 with retention time 12.28 min and Intensity 60199 and two with retention time 12.82 and I=132469) -> the impression that a reader has is that this duplication is intentionally done to achieve a total of 113 compounds in the extracts.

Then there are the cases of compounds which differ for MW, atomic composition and retention times, but are all named with the same common name and CAS number (ex. 7-hydroxycumarin).

A revision of the original data with a proper comparison of the compounds is mandatory to rationalize the entire study.

Table 2 serves to visually show the different levels of common compounds, but only 2 are colour coded, as such it is incomplete therefore useless.

Regarding the biological effect, the extracts have been used as such before fractionation, therefore many of the effects might be due to more than 1 specific compound. The non toxicity of the extracts towards Escherichia coli is partially convincing. In laboratory condition a strain grown in LB (a rich medium) after 24h has reached at least 2 unity of Abs, not 0.15. This is a value after 2 h not 24 h. In comparison with ampicillin, however, it is clear that the bacteriostatic effect is negligible.

From the data in Figure 1 it is evident that the nematocidal activity is higher for the J.o. extracts, which are comparable with the nocodazole effects. This figure is indeed composed of 4 panels, not 2 because it represents the results of 4 different types of experiments, therefore the legend should be changed accordingly. Moreover, there is a problem with the 4 graphs reporting the % of adult development as a function of the molecules of extracts in the incubation medium. The value of the nocodazole's effect should always be the same, unless different concentration of the drug are used. Given the lack in the legend, it is not possible to understand the reason for this discrepancy.

Figure 2E is puzzling. After four days even the control has a total drop in the brood size, so the graph on the right is not the best way of presenting the data, a direct comparison of the size per day is pertinent and therefore only the difference between the control and the S.l. extract is significant.

In Figure 2D the authors claim that there are no chromatin bridges in the micrographs, however the resolution is too low to demonstrate this sentence. The total fluorescence is lower in the treated sample than in the control, but nothing more can be deduced.

In figure 3A the herbal extract of S.l. downregulates atm gene, it does not activate it, differently from what written in the legend. In Figure 3B pCHK-1 should be shown in the red channel, but nothing is written in the method section regarding the fluorophore used in the immunostaining. Moreover, no red points/puncta are visible next to the arrows. Figure 3C is puzzling because for the data reporting the effect of S.l. extracts, there are two sets of data: 5 points at the same values as the control and 5 points at high apoptosis ratio. This is not at all taken into account nor commented by the authors.

Paragraph 2.9: the deduced cause of the nematocidal activity of the three extracts is not ROS alone, since the treatment with N-AcylCys does not restore the vitality. Moreover the data in figure 1A and Figure 5 without the NAC should be comparable, but it is not written at which concentration the extracts were used. One of the three values used in Figure 1 should be selected for the experiment in Figure 5, which in the main text is cited as Figure 6.

Figure 6 is a graphical abstract more than an extra figure and the real shared compounds are not 113, but about 60.

Minor points: please, take care of putting in italics all the species names (ex. lines 66, 70, 99); do not end each paragraph of the results with a conclusion sentence, which is then repeated in the discussion. Sentence in lines 167-169 lacks a verb, please rephrase it. Line 171 -> herbal extract not "her extract". Line 363 -> ROS, not "OS".

Reviewer 2 Report

Comments and Suggestions for Authors

In the MS nutrients-3903433, the authors aim to attract the readers' interest by investigating the nematocidal activity of 3 plant extracts (Ruscus hyrcanus, Juniperus oblonga, and Stachys lavandulifolia). 

The following comments are available below:

The article is based on 42 references from 1979 to 2024; 7 are self-citations (14, 16, 18, 19, 21, 23, 40), and only 12 were published in the last 5 years. The authors are invited to organize all references in the MDPI style, verify their relevance, and diminish the number of self-citations.

1. Abstract: 

The authors are invited to support the narrated results with some statistically significant values. Moreover, the conclusions should be strictly related to the results. The authors are invited to verify the keywords, maintaining only the most relevant ones.

2. Introduction:

The authors are invited to present the nutritional values of all three investigated plant species, as well as the well-known bioactive compounds. Then, they are encouraged to show the traditional mode of their consumption, as well as the ways used to administer them and achieve their therapeutic effects.

Please provide more recent data from the literature on the nematocidal activity of the studied species and formulate some hypotheses.

Finally, the authors are invited to show the aim of their study and its novelty. 

3. Materials and Methods:

The authors are invited to include a first subsection entitled "Materials," which should contain the following: plant materials (the plant parts harvested, the GPS coordinates of the harvesting places, the period of harvesting process), solvents used for extraction, standards used for LC-MS, bacterial cell lines, C. elegans used for in vivo studies, reagents, various kits, and their provenance.

The references are missing in 3.4. and 3.8. 

The authors are invited to provide the name and principle of each method used in the present study, and briefly describe the working conditions, controls, and samples, along with the tested concentrations, supported by suitable references. Thus, the reader can understand and appreciate the accuracy and reproducibility of the methods used. 

Please organize the Materials and Methods presentation as follows:

3.1. Materials

3.2. Plant extracts' preparation, mentioning all details, and the preservation conditions.

3.3. LC-MS Analysis with aparathus, conditions, standards, etc.

3.4. Antibacterial activity - please mention the method used, controls, and samples, and the time (24, 48 hours) used for incubation. 

3.5. In vivo studies on C. elegans

3.6. Statistical Analysis: Please mention the software and the methods used to support the obtained results.

4. Results:

This section should contain only the original results, without references.

In the current manuscript, the authors have presented the results in an inappropriate manner, currently used in the Discussion section. Therefore, the reader can not differentiate the original findings from the mixed data to appreciate the accuracy of the original results. In the attached manuscript, the reviewer marked with light yellow the paragraphs that should be included in the Discussion section. 

The authors should maintain a similar sequence and subsection names as in the Materials and Methods section. Thus, the reader can appreciate the value of the original results and easily correlate them with Materials and Methods. 

Subsection 2.1. This subsection should contain the yield of each extract obtained.

Subsection 2.2. It should be the LC-MS analysis. In the current version, the original results are not presented appropriately. The authors are invited to include the LC-MS chromatograms for each extract and identify the most significant compounds in each figure. 

They should include a table to present the concentrations (mean ± SD) of the most abundant phytochemicals in each extract in a comparative manner, indicating the statistical differences by calculating p-values. The current Figure 4 and Table 1 are not relevant. Both can be included in the Supplementary material to serve as a base for the Discussion section. 

Subsection 2.3. Antibacterial activity

The results (mean ± SD) should be presented in a table, considering each extract, controls, and conventional antibiotics.  The current figure (1B) is difficult for the reader to decipher, and the y-axis has no name. 

Subsection 2.4. In vivo studies on C. elegans

The authors are invited to present all results (expressed as mean ± SD) in corresponding tables, for each extract, along with controls, including the calculated p-values. The authors are invited to verify the measurement units for all measured parameters and mention them at the head of each column.

All statistical graphs should be included in the supplementary material. 

The most relevant images could be included in the MS text, verifying their resolution, quality, and necessary notations for readers, thereby facilitating their understanding of the obtained results. 

5. Discussions:

First, the authors must justify the present study design, including the plant species tested, the plant parts harvested, the solvents used, and the extraction method employed. 

Then, the LC-MS results should be interpreted in comparison to those of other studies and the differences explained. 

The same request is available for all obtained results. 

Moreover, the authors should statistically correlate the chemical constituents of each extract with its pharmacological effects to support the original results. Then, they must discuss all correlations, comparing them with those from the literature data. 

6. Conclusions:

After all requested revisions, the authors should reformulate the conclusions, showing the most relevant data from the present study. Then, they can indicate further directions of the present research. 

Finally, the authors are invited to display all abbreviations and their explanations in a table, at the end of the manuscript. 

Comments on the Quality of English Language

The English and editing style are cumbersome. All abbreviations should be verified and consistently maintained throughout the manuscript. Numerous misprints should be corrected. 

Round 2

Reviewer 1 Report

Comments and Suggestions for Authors

The manuscript has been revised and the data have been more carefully analysed, as such it is now acceptable for publicaiton

Author Response

We sincerely thank Reviewer 1 for the careful and thorough evaluation of our manuscript. We greatly appreciate your insightful comments and suggestions, which helped us improve the clarity and quality of our work.

Reviewer 2 Report

Comments and Suggestions for Authors

The reviewer appreciates the authors' efforts to revise the manuscript according to the comments from the previous review report. 

However, they partially responded to the Reviewer comments, and the following observations are still available:

Abstract:

In the revised version, it is formed by objectives, results, and conclusions. Materials and methods are missing. In the results, the requested statistically significant values are missing, too. Three different plant extracts were analyzed; the differences between them are expected to be evidenced and supported by statistically significant values. The LS-MS results can provide the classes of natural compounds identified (several examples).

Results: 

The reviewer still considers that all titles from the Results section should correspond to those from Materials and Methods. Therefore, the reader can easily make a clear correspondence between the technique used and the results obtained.  In the discussion section, the authors are free to use more relevant titles for the corresponding subsections to highlight the value of their original results. 

The following comments contain some examples with doubtful subsection and sub-subsection titles. Therefore, the authors are encouraged to revise all of them. 

The relevance of Table 1 can be enhanced if the authors display all representatives identified by LC-MS for each class. The reviewer also suggests relocating Figure 1 to the Supplementary Material, as all subfigures repeat the same data, and their visibility is low.

The sub-subsection 222 (lines 108-142) should be included in the Discussion section. Moreover, the title is incorrect, as the authors did not present the results of the nematocidal activity evaluation here. Please write the scientific names in italics. 

Table 2 is irrelevant, because all three columns contain the same data.

The authors can remove it, and specify in the MS text, in the discussion sections, the main classes and their representatives. If they want to maintain it, only the first two columns are enough for the reader to understand the table caption, which states that all 13 phytochemicals are found in all three plant extracts.

The authors are invited to reduce the Table caption, including only the most relevant words. These words generally indicate the data from the table, and not repeat the information contained in it. Moreover, the Table caption should be placed above the table. All other details can be included in the manuscript text. 

Transfering subsection 2.2.2. in the Discussion section, as requested, the sub-subsection 2.2.1. is not necessary. 

For better understanding, the title of Subsection 2.3. (line 144) must be revised, specifying, for example, Antibacterial Activity on E. coli OD50. 

The current version, "Three herb Extracts do not alter Bacterial Growth: Antibacterial activity," is: 

  • incorrectly grammatically written
  • too general, because the antibacterial activity is evaluated only on E coli OD50, a strain conventionally used as a bacterial food in the laboratory maintenance of C. elegans on agar plates.

Figure 2. Please correct: absorbance

2.4. Nematocidal Effects

Figure 3. Please explain what the symbol * means. Moreover, when presenting comparative values in the manuscript text, please include the p-values (p<0.05 or p>0.05) to indicate whether the differences are statistically significant.

The same request is available for Subsection 2.5. 

Please write the p-value constantly in the MS text. Usually, it is edited with italics p (as in lines 322-325); in line 285 is "P" and in line 415 is "p."

Discussions:

The phrases from lines 457-460 are repeated. Please mix it with that from the Materials and Methods, resulting in a complete description of plant material harvesting and identification. 

The reviewer expected to show why these parts were selected for the present study, and not only leaves or only fruits. 

Please organize the Discussion Section as follows:

  • A brief presentation of the study design, with all phases, showing the reason for their selection. Why did the authors select C. elegans and not other nematode species for evaluating the nematocidal effect? 
  • This should be followed by all discussions and presented in the same order as in the previous sections (Materials and Methods and Results).

In this section, the authors could benefit from using more relevant titles for the subsections to more effectively highlight their valuable results. 

In the present study, the authors tested the plant extracts, and the abbreviation of plant extracts usually includes E. For example, RHE, SLE, and JOE are suitable abbreviations for the present study. The abbreviations used in the current manuscript for the plant scientific names are uncommon: R.h, S.l, J.o. Moreover, in the entire MS text, they must specify the plant extracts, obtained through the described technique, not only R.h, S.l, and J.o.

Comments on the Quality of English Language

Moderate revision. 

Numerous misprints and ambiguous expressions can be found in the text of the manuscript. Partially, they were illustrated in the Comments for authors. 

In the present study, the authors tested the plant extracts, and the abbreviation of plant extracts usually includes E. For example, RHE, SLE, and JOE are suitable abbreviations for the present study. The abbreviations used in the current manuscript for the plant scientific names are uncommon: R.h, S.l, J.o. Moreover, in the entire MS text, they must specify the plant extracts, obtained through the described technique, not only R.h, S.l, and J.o.

Author Response

Reviewer 2)

The reviewer appreciates the authors' efforts to revise the manuscript according to the comments from the previous review report. 

However, they partially responded to the Reviewer comments, and the following observations are still available:

Abstract:

In the revised version, it is formed by objectives, results, and conclusions. Materials and methods are missing. In the results, the requested statistically significant values are missing, too. Three different plant extracts were analyzed; the differences between them are expected to be evidenced and supported by statistically significant values. The LS-MS results can provide the classes of natural compounds identified (several examples).

Response:
We thank the reviewer for this constructive comment. In the revised Abstract, we have now briefly included the experimental approach to clarify the materials and methods used. We also added information on the classes of compounds identified by LC–MS (“mainly including phenolic acids, flavonoids, and terpenoids”) and indicated statistical significance in the results section (“significantly (p < 0.05) reduces survival…”). These revisions address the reviewer’s concerns while keeping the Abstract concise and focused.

Line 13-

 Synchronized young adult worms were exposed to each extract, and survival and reproductive parameters were statistically analyzed using two-tailed Mann–Whitney tests. Through LC–MS analysis, we identified that all three extracts shared 78 compounds, mainly including phenolic acids, flavonoids, and terpenoids. Our findings indicate that reactive oxygen species generation is a major driver of nematocidal and fertility-reducing effects, while modulation of DNA damage response pathways further contributes to germline defects. Taken together, these results demonstrate that exposure to the extracts significantly (p< 0.05) reduces survival, impairs larval development, elevates the High Incidence of Males phenotype, and disrupts germline integrity in a dose-dependent manner.

Results: 

The reviewer still considers that all titles from the Results section should correspond to those from Materials and Methods. Therefore, the reader can easily make a clear correspondence between the technique used and the results obtained.  In the discussion section, the authors are free to use more relevant titles for the corresponding subsections to highlight the value of their original results. 

Response:
We appreciate the reviewer’s helpful suggestion. Accordingly, we have revised all subsection titles in the Results section to match those used in Materials and Methods. (Results 2.1- 2.9 and Materials and methods 3.1 to 3.9). This alignment ensures that each experimental approach.  

The following comments contain some examples with doubtful subsection and sub-subsection titles. Therefore, the authors are encouraged to revise all of them. 

The relevance of Table 1 can be enhanced if the authors display all representatives identified by LC-MS for each class. The reviewer also suggests relocating Figure 1 to the Supplementary Material, as all subfigures repeat the same data, and their visibility is low.

We appreciate the reviewer’s helpful suggestion. To address the current suggestion, Table 1 now displays all representative compounds identified by LC–MS for each chemical class.

For Figure 4, During the first revision, we relocated this panel from the original Figure 4 to its current position per the reviewer’s request, so that readers encounter the chemical-profile data prior to the biological-results section. This reordering required renumbering all subsequent figures to maintain logical flow.

To balance clarity and the reviewer’s request, we moved subpanel 1A to Supplementary Figure S1 while retaining Figure 1B to preserve logical flow and data accessibility. This approach preserves the logical sequence of the main text and biological results, avoids extensive renumbering, and ensures that essential LC–MS data remain accessible to the readers. We hope the reviewer finds this compromise reasonable and appreciate the guidance to improve clarity and readability.

The sub-subsection 222 (lines 108-142) should be included in the Discussion section. Moreover, the title is incorrect, as the authors did not present the results of the nematocidal activity evaluation here. Please write the scientific names in italics. 

Table 2 is irrelevant, because all three columns contain the same data.

The authors can remove it, and specify in the MS text, in the discussion sections, the main classes and their representatives. If they want to maintain it, only the first two columns are enough for the reader to understand the table caption, which states that all 13 phytochemicals are found in all three plant extracts.

As suggested, Table 2 has been removed from the text.

Also, As suggested, subsection 2.2.2 has been removed since its content has been integrated into the revised Discussion (Section 4.1), which now includes the interpretation of LC–MS results and the description of the 13 representative bioactive compounds.

The authors are invited to reduce the Table caption, including only the most relevant words. These words generally indicate the data from the table, and not repeat the information contained in it. Moreover, the Table caption should be placed above the table. All other details can be included in the manuscript text. 

Transfering subsection 2.2.2. in the Discussion section, as requested, the sub-subsection 2.2.1. is not necessary. 

Response:
We thank the reviewer for this suggestion. We have moved sub-subsection 2.2.2 to the Discussion section, as recommended. Additionally, the title has been revised to “Classification of Representative Phytochemicals and Their Reported Bioactivities” to accurately reflect the content, focusing on LC–MS-identified compounds and their reported bioactivities rather than nematocidal activity evaluation. All scientific names have been formatted in italics.

Line 105, Table caption is now placed above of the table (Table 1)

For better understanding, the title of Subsection 2.3. (line 144) must be revised, specifying, for example, Antibacterial Activity on E. coli OD50. 

The current version, "Three herb Extracts do not alter Bacterial Growth: Antibacterial activity," is: 

  • incorrectly grammatically written
  • too general, because the antibacterial activity is evaluated only on E coli OD50, a strain conventionally used as a bacterial food in the laboratory maintenance of C. elegans on agar plates.

Response:. The title of Subsection 2.3 has been revised to “Antibacterial Activity of Three Herb Extracts on E. coli OP50” to accurately reflect the specific assay performed (Line 115).

Figure 2. Please correct: absorbance

Response:
The text has been revised (Figure 2)

2.4. Nematocidal Effects

Figure 3. Please explain what the symbol * means. Moreover, when presenting comparative values in the manuscript text, please include the p-values (p<0.05 or p>0.05) to indicate whether the differences are statistically significant.

The same request is available for Subsection 2.5. 

Please write the p-value constantly in the MS text. Usually, it is edited with italics p (as in lines 322-325); in line 285 is "P" and in line 415 is "p."

Response:
We thank the reviewer for the suggestion. The meaning of the symbol * in Figure 3 has been clarified in the figure legend: * indicates p < 0.05 compared with control.

The text in Section 2.5 has been revised to include p-values for all comparative survival values, and the p-value notation has been standardized to lowercase italics throughout the manuscript. Specifically, the revised sentence now reads:

Line 173-

For example, in R.h-treated worms, survival decreased from 60.3% to 57.2% and 55.1% at concentrations of 0.03, 0.3, and 3 µg/ml, respectively (p < 0.05 compared with control). Similarly, survival rates for J.o declined from 38.5% to 35.5% and 33.4% (p < 0.05), while those for S.l decreased from 52.9% to 49.8% and 47.7% (p < 0.05) across the same concentration range

Discussions:

The phrases from lines 457-460 are repeated. Please mix it with that from the Materials and Methods, resulting in a complete description of plant material harvesting and identification. 

The reviewer expected to show why these parts were selected for the present study, and not only leaves or only fruits. 

Please organize the Discussion Section as follows:

  • A brief presentation of the study design, with all phases, showing the reason for their selection. Why did the authors select C. elegans and not other nematode species for evaluating the nematocidal effect? 

The above reviewer’s comments were addressed as follows. The section has been revised to read:

Line 438

In our previous screening of over 316 herb extracts, ~16% caused reduced survival, larval arrest, or lethality in C. elegans [29]. Three extracts showing the most consistent effects were selected for focused study. C. elegans was chosen for its well-characterized development, short lifecycle, fully sequenced genome, and reproducible endpoints, allowing rapid, high-throughput evaluation of nematocidal effects. Its conserved stress response and ROS-mediated pathways also make it suitable for investigating herb-induced cytotoxicity.

Line 510

The choice of plant parts and solvent fractions was not predetermined but guided by preliminary C. elegans assays, which identified the n-butanol extracts as those showing the most reproducible bioactivity among the tested fractions.

  • This should be followed by all discussions and presented in the same order as in the previous sections (Materials and Methods and Results).

In this section, the authors could benefit from using more relevant titles for the subsections to more effectively highlight their valuable results. 

 Response) We accepted the reviewer’s suggestion and reordered the Discussion accordingly.

In the present study, the authors tested the plant extracts, and the abbreviation of plant extracts usually includes E. For example, RHE, SLE, and JOE are suitable abbreviations for the present study. The abbreviations used in the current manuscript for the plant scientific names are uncommon: R.h, S.l, J.o. Moreover, in the entire MS text, they must specify the plant extracts, obtained through the described technique, not only R.h, S.l, and J.o.

Response:
We thank the reviewer for the comment. The manuscript has been revised to standardize the abbreviations of plant extracts as RHE, SLE, and JOE throughout the text, including all figure legends. It is also clearly indicated that all experiments were conducted using the n-butanol fraction of each extract (line 81-85).

Example revision in the manuscript (Line 89–92):

All three n-butanol fractions of Ruscus hyrcanus extract (RHE), Juniperus oblonga ex-tract (JOE), and Stachys lavandulifolia extract (SLE) revealed 79 compounds, 78 of which were common, although their relative abundances varied among the extracts (Figure 1A and Supplementary Tables S1–S4).

All the figure legends have been reflected changes